# PeerJ

# Microbial secondary succession in soil microcosms of a desert oasis in the Cuatro Cienegas Basin, Mexico

Nguyen E. López-Lozano[1], Karla B. Heidelberg[2], William C. Nelson[2], Felipe García-Oliva[3], Luis E. Eguiarte[1] and Valeria Souza[1]

[1] Departamento de Ecología Evolutiva, Instituto de Ecología, Universidad Nacional Autónoma de México, México
[2] Department of Biology, University of Southern California, Los Angeles, CA, USA
[3] Centro de Investigaciones en Ecosistemas, Universidad Nacional Autónoma de México, Morelia, México

## ABSTRACT

Ecological succession is one of the most important concepts in ecology. However for microbial community succession, there is a lack of a solid theoretical framework regarding succession in microorganisms. This is in part due to microbial community complexity and plasticity but also because little is known about temporal patterns of microbial community shifts in different kinds of ecosystems, including arid soils. The Cuatro Cienegas Basin (CCB) in Coahuila, Mexico, is an arid zone with high diversity and endemisms that has recently been threatened by aquifer overexploitation. The gypsum-based soil system of the CCB is one of the most oligotrophic places in the world. We undertook a comparative 16S rRNA 454 pyrosequencing study to evaluate microbial community succession and recovery over a year after disturbance at two sites. Results were related to concurrent measurements of humidity, organic matter and total C and N content. While each site differed in both biogeochemistry and biodiversity, both present similar pattern of change at the beginning of the succession that diverged in later stages. After one year, experimentally disturbed soil was not similar to established and undisturbed adjacent soil communities indicating recovery and succession in disturbed soils is a long process.

## INTRODUCTION

In ecological theory, succession is defined as the predictable manner by which communities change over time during the colonization of a new environment or following a disturbance (primary and secondary succession respectively) (*Begon, Townsend & Harper, 2006*). Bacterial succession has been explored in a variety of environments and over different timescales (*Fierer et al., 2010*). However, there are many theoretical and methodological questions that remain unanswered. For example the order of occurrence and species turnover in successional stages can differ and be explained by stochastic or functional factors depending on the use of ecological or functional classifications

Corresponding author
Valeria Souza,
souza.valeria2@gmail.com

instead of 16S rRNA taxonomic classifications (*Burke et al., 2011a*; *Burke et al., 2011b*). Notwithstanding, there are theoretical models for microbial community succession and assembly, most of them derived from those already well defined for macroorganisms (*Prosser et al., 2007*). Due to the complexity of the patterns observed in the succession of microbial communities, alternative explanations started to appear based on carbon (C) inputs to the systems, suggesting that species richness and the biomass of specific ecological groups change along successional stages (*Fierer et al., 2010*).

In recent years, the role of deterministic and stochastic processes in community assembly has been tested. It has been shown that bacterial community assemblages can be regulated by the local environment (species sorting) (*Langenheder & Székely, 2011*), massive immigration that can prevent competitive exclusion of species (mass effect) (*Lindström & Langenheder, 2012*), and neutral process, which assumes that all species are similar in their competitive ability and in dispersal. In the neutral process scenario, changes in community composition result from neutral drift over time (*Woodcock et al., 2007*; *Ofiteru et al., 2010*). These mechanisms can co-occur, resulting in microbial communities being structured by more than one process (*Langenheder & Székely, 2011*). However, the development of such theoretical models are based mostly on work done in laboratory environments, and little is known of what happens in natural systems such as soils (*Caruso et al., 2011*; *Navarro et al., 2009*; *Nemergut et al., 2007*; *Schmidt, Costello & Nemergut, 2007*). Data on how microbial communities change through time in natural environments are needed in order to refine these theoretical models.

Soils are among the most diverse microbial environments analyzed to date (*Youssef & Elshahed, 2009*; *Daniel, 2005*), which makes the identification of differences in community diversity patterns between stable and disturbed soils challenging. Arid soils are thought to have lower productivity and diversity than other soil habitats, and therefore may provide a better opportunity to validate molecular ecosystems approaches to examine the genetic and functional organization of native microbial consortia. Earth's arid regions represent today nearly one third of total continental ecosystems (*Collins et al., 2008*) and are thought to be more vulnerable under most scenarios of global climate change (Intergovernmental Panel on Climate Change, 2007 report).

The Cuatro Cienegas Basin (CCB) in Coahuila, Mexico, is an arid zone that has recently been threatened by aquifer overexploitation. The gypsum-based soil system at CCB is one of the most oligotrophic environments in the world despite the high diversity in comparison with other arid soils (*López-Lozano et al., 2012*). Nevertheless, due to recent and ongoing overexploitation of the deep aquifer in this oasis, we decided to evaluate the community sensitivity after disturbance. In this study we experimentally sampled and disturbed soil microcosms with gamma rays and sterilization. Samples were restored to their original site in a mesh bag that allowed migration. Then, community composition as well as nutrient dynamics were followed and compared to neighboring undisturbed soil samples. The objective of this study was to evaluate the recovery of microbial biodiversity and biogeochemical characteristics of the Churince within the CCB during a process of microbial secondary succession over one year period after disturbance and to find out if

microbial succession the shift pattern behaves like any of the previously proposed models. Our approach was designed to be experimentally analogous to sections of a forest cleared by removing the biota of an area, and tracking recolonization by migration.

## MATERIALS AND METHODS

*Site description and experimental design.* The sampling sites are located at the Churince oasis system on the west side of the Cuatro Cienegas Basin (CCB) at 740 m above sea level. This valley has <15.0 cm of annual precipitation falling mainly during the summer. The dominant soil is gypsisol, and the predominant vegetation types are halophile and gypshophile grasses. As described before (*López-Lozano et al., 2012*), the Churince system consists of a spring, an intermediate lagoon, and a desiccation lagoon (Dry Lagoon), connected by short shallow creeks. Two sites separated by one kilometer were selected. One site borders a small river creek (River) (26°50′43″N/102°8′18″W), and has 60% of plant coverage. The grass *Sporobolus airoides* is the dominant plant. The other site is closer to a dry lagoon (26°50′53″N/102°8′52″W) and has only 10% of plant coverage. At this site the gypsophile grass, *Sesuvium erectum,* is the dominant plant species. The sites have alkaline soils with slight variation in the pH of 8.5–8.8.

In January of 2007, 20 kg of soil were collected from each site. The soil was mixed and used to construct microcosms of 1 kg of soil in permeable bags of nylon mesh (25 × 25 cm). The microcosms were sterilized by autoclaving followed by a dose of 25 kGy of gamma rays (at the Instituto Nacional de Investigaciones Nucleares, La Marquesa, México). One microcosm sample from each site was immediately used for DNA extraction as a control.

In February 2007, plots of 8 × 8 m were established at each site. The plots were divided in 64 quadrants of 1 m$^2$, and 40 microcosm bags were placed randomly within the plots as replicates. Every three months, three microcosm bags were collected randomly during a period of one year (3, 6, 9 and 12 months), 12 in total for each site. The remaining microcosms were deposited in the site in order to guarantee enough samples in each sampling period but not analyzed for this experiment. In addition, three samples from 15 cm$^3$ of undisturbed soil next to the microcosms in each plot were randomly collected at the beginning and end (12 months) of the experiment to describe the community that potentially could colonize the microcosms. 50 g of each sample were stored in liquid nitrogen until DNA extraction. The remaining soil was stored in black plastic bags at 4 °C during one month until processing in the laboratory for biogeochemical analysis.

*Soil biogeochemical analyses.* Separate samples were oven-dried at 75 °C to constant weight for soil moisture determination by the gravimetric method (*Reynolds, 1970*). Measurements of (C) forms were determined with a Total Carbon Analyzer (UIC Mod. CM5012), while nitrogen (N) and phosphorus (P) forms were determined using a Bran-Luebbe Auto Analyzer III (Norderstedt, Germany). Total and inorganic C were determined by dry combustion and coulometric detection (*Huffman, 1977*). Organic C was calculated as the difference between total and inorganic C. N and P concentrations were determined following acid digestion; N was determined using a modified Kjeldahl method

(*Bremmer & Mulvaney, 1982*), and P was determined with the molybdate colorimetric method following ascorbic acid reduction (*Murphy & Riley, 1962*). Microbial C (micC) and microbial N (micN) concentrations were determined from field-moist samples by a chloroform fumigation extraction method (*Vance, Brookes & Jenkinson, 1987*).

Inorganic N ($NH_4^+$ and $NO_3^-$) was extracted from fresh sub-samples with 2 M KCl, followed by filtration through a Whatman No. 1 paper filter (*Robertson et al., 1999*), and determined colorimetrically by the phenol-hypochlorite method. Dissolved organic C, N and P were extracted in two grams of fresh material with deionized water after shaking for 1 h, and filtering through a Whatman #42 filter and a 0.45 μm nitrocellulose membrane (*Jones & Willett, 2006*). Dissolved organic C (DOC) was determined by combustion and coulometric detection (*Huffman, 1977*). Dissolved organic N (DON) and dissolved organic P (DOP) were determined after acid digestion. DON was calculated as the difference between digested soluble N and $NH_4^+$ in deionized water extracts. The DOP was calculated as the difference between digested dissolved P and inorganic P (as orthophosphate).

*Statistical analyses.* To evaluate significant differences in biogeochemical soil parameters between sites and through the time during the microcosms experiment, we analyzed biogeochemical variables with a repeated measures analysis of variance (RMANOVA) with one between-subject factor (site: Dry Lagoon and River) and one within factor (sampling date: 3, 6, 9 and 12 months). In order to compare biogeochemical soil parameters between sites, and between disturbed and undisturbed soil in the last sampling date (12 months), we carried out a two-way analysis of variance (ANOVA) (factor 1 levels: sites Dry Lagoon and River; factor 2 levels: undisturbed soil and microcosm).

The relationship between microbial community composition in terms of the most abundant bacterial families, soil characteristics and samples (date and site) was analyzed by canonical correspondence analysis (CCA). In this analysis, the samples are represented by a centroid. Its position is indicative of the relationship between a specific sample and either of the ordination axes. Soil characteristics are represented by vectors. Vectors of greater magnitude and forming smaller angles with an ordination axis are more strongly correlated with that ordination axis. High scores of absolute value for a given family or a given site on a CCA axis indicate that it is highly related to the axis and to the environmental variable exhibiting high correlation to the axis. All soil characteristics were tested for significant contribution to the explanation of the variation in bacterial family community composition with an ANOVA like permutation test to assess the significance of constraints. Only variables that were significant by the permutation test at the $P \leq 0.05$ level were included in the CCA biplot. The CCA was performed using the package Vegan in R (http://www.r-project.org/). In the supplementary material, we added a DCA analysis without the restrictive variables for comparison (Figure S2).

*DNA Extraction.* Total DNA was extracted using the Soil Master DNA Extraction Kit (EPICENTRE Biotechnology) according to the manufacturer's instructions, with an additional step of bacterial isolation using the fractionation centrifugation technique described in *Holben et al. (1988)*. This step was performed on frozen 50 g soil samples

before DNA extraction as a way of reducing the remaining concentrations of salts, polysaccharides and secondary compounds. DNA was stored at $-20\,°C$.

In order to confirm sterilization in the control microcosms, the DNA region coding for 16S rRNA was PCR amplified using universal primers (Tables S1 and S2). Since no band was recovered in a PCR gel run using the sterilized soil, the sterilization was assumed to be effective.

*Pyrosequencing of 16S rRNA tags.* 16S rRNA genes were amplified from each sampling date (pooled DNA from the three subsamples was used as template) using the 939F (TTGACGGGGGGCCCGCAC) and 1492R (TACCTTGTTACGACTT) paired primers. Sequencing was undertaken using the standard Roche 454 Titanium LIB-A kit with multiplex identifier sequence (MID) tags (Table S3) (*Sun, Wolcott & Dowd, 2011*) at the Research and Testing Laboratory (Lubbock, TX).

*Bioinformatic analysis of barcoded 16S ribosomal RNA gene libraries:* Mothur open-source software package (v 1.15.0) (*Schloss et al., 2009*) was used for processing, clustering, and classification of the quality screened sequence data. Raw sequences were screened for potential chimeric reads using Chimera.slayer (*Haas et al., 2011*) and the linked SILVA template database (26%). Sequences containing homopolymer runs of 9 or more bases, those with more than one mismatch to the sequencing primer and those longer than 565 bp were eliminated. Group membership was determined prior to trimming of the bar-code MID and primer sequence. Sequences were aligned against the manually curated bacterial SILVA 16S rRNA gene template using the nearest alignment space termination (NAST) algorithm (*Schloss, 2010*; *DeSantis et al., 2006*), and manually trimmed for the optimal alignment region (start $= 28596$, end $= 38347$), which yielded aligned fragments of 270 bp long, including both the V6 and V7 regions. Pre-clustering, based on the SLP clustering algorithm (*Huse et al., 2010*), was used to reduce the effect of pyrosequencing errors on subsequent analysis. A pairwise distance matrix was calculate across the non-redundant sequence set, and reads were clustered into operational taxonomic units (OTUs) at 3% distance using the furthest neighbor method (*Schloss & Handelsman, 2006*). This matrix was used to calculate the similarity based on the Bray-Curtis and Jaccard Index between the samples in each site and to create a cluster diagram of sample similarity. The sequences and OTUs were categorized taxonomically using Mothur's Bayesian classifier and the SILVA bacterial reference set. A representative sequence from the center of each OTU grouping was classified using a naïve Bayesian approach (*Wang et al., 2007*). The taxonomic reference database was composed of unique, full-length sequences from the SILVA SSU Ref v.106 database (http://www.mothur.org/wiki/Silva_reference_files) (*Pruesse et al., 2007*). To account for the effects of different sequencing depths on the alpha-diversity measurements, the sample sets were normalized to equal abundance of the sample with the least sequencing effort (Dry Lagoon, 9 months; 10,441 sequences; Table S3). Alpha-diversity statistics including Chao1 non-parametric species richness estimate (calculates the estimated true species diversity of a sample based in the number of singletons and doubletons), Shannon indices, Shannon-based richness estimate (diversity indices as a quantitative measure that reflects how many different OTUs there are in the

dataset, simultaneously takes into account how evenly the OTUs are distributed), and Good's coverage estimate (percent of the total OTUs represented in the sample), were generated for each sample using the same program.

The GenBank accession numbers for 16S rRNA pyrosequences in this work are in the BioProject: PRJNA167137. The sample identifications for microcosms in the Dry Lagoon are: before sterilization SRS346170, at 3 months SRS346171, at 6 months SRS346172, at 9 months SRS346173 and at 12 months SRS346174. The samples ID for microcosms in the River are: before sterilization SRS346176, at 3 months SRS346177, at 6 months SRS346178, at 9 months SRS346179 and at 12 months SRS346180. The samples ID for the undisturbed adjacent soil at 12 months are SRS346175 for the Dry Lagoon, and SRS346181 for the River.

## RESULTS

*Nutrient fluctuations during the secondary succession.* In order to understand the relationships between the changes in community composition and nutrient transformations that occurred during the secondary succession process in the Churince soils, the soil samples were biogeochemically characterized and compared. We found that the CCB soil at the Dry Lagoon site was more oligotrophic (C:N:P ratio of 125:5:1) than the River site (C:N:P ratio 300:16:1) and in comparison with a general "average" soil C:N:P of 186:13:1 (*Cleveland & Liptzin, 2007*). RMANOVA analysis revealed consistent differences in the soil nutrients between sites across time (Table 1). The Total Organic C was higher at the site adjacent to the River throughout the sampling, even though it showed a fluctuating pattern during the secondary succession experiment, probably due to both seasonal effects and succession stages (Table 1). The same pattern was observed for Total N and Total P. On the other hand, microbial C (micC) as an indicator of microbial biomass showed similar fluctuations in both sites (Table 1). Values increased during the first three and six months after sterilization, then micC decreased at 9 months and increased again in the last sampling date. Ammonium showed contrasting patterns between sites. At the Dry Lagoon, ammonium increased showing higher values at 6 months, and then decreased gradually. At the River, ammonium increased gradually to drop in the last sampling. The microbial N (micN) increased in samples near the River, but presented fluctuations in the Dry Lagoon. Only the nitrate was higher in the Dry Lagoon microcosms but there were not significant differences by time. ANOVA analysis of soil nutrients in microcosms samples and undisturbed adjacent soil at 12 months showed significant differences between treatments in Total Organic C, dissolved organic N (DON), dissolved organic P (DOP) and ammonium (Table 2). However, undisturbed soil samples were similar to the perturbed samples in their total N and P content (TN, TP and DIP) (Table 2). These data suggests that, in general, the sampling sites are different and as such, respond differently to perturbation.

*Evaluation of community structure by 16S rRNA amplicons.* The number of high quality sequences per library ranged from 10,441 to 41,074 per sample (Table S3). Once the dataset was normalized to the sample with the lowest sequencing effort, 79,303 OTUs

**Table 1 Soil and nutrients.** Soil biogeochemical parameters in each sampling date for both sites Dry Lagoon and River. Analysis was done as a 2 way ANOVA, the mean ± SD with $F$ values and significance levels.

| | Dry Lagoon | | | | | River | | | | | Effect | | |
|---|---|---|---|---|---|---|---|---|---|---|---|---|---|
| | | | | | | | | | | | Site | Time | site*time |
| | | | | | | | | | | | F values | | |
| | Sterilized soil | 3 months | 6 months | 9 months | 12 months | Sterilized soil | 3 months | 6 months | 9 months | 12 months | | | |
| TOC ($mg\,g^{-1}$) | 0.3 ± 0.13 | 1.8 ± 0.60 | 2.5 ± 0.03 | 3.9 ± 0.30 | 2.9 ± 0.20 | 6.5 ± 0.60 | 9.8 ± 0.86 | 7 ± 1.10 | 15 ± 2.10 | 4 ± 1.70 | 61.4*** | 10.68*** | 7.67** |
| TN ($mg\,g^{-1}$) | 0.15 ± 0.01 | 0.16 ± 0.06 | 0.13 ± 0.004 | 0.23 ± 0.01 | 0.24 ± 0.02 | 0.8 ± 0.01 | 0.6 ± 0.08 | 0.6 ± 0.03 | 0.8 ± 0.04 | 0.7 ± 0.06 | 226.9*** | 3.58* | 0.77ns |
| TP ($mg\,g^{-1}$) | 0.02 ± 0.001 | 0.01 ± 0.002 | 0.01 ± 0.001 | 0.05 ± 0.002 | 0.02 ± 0.001 | 0.04 ± 0.006 | 0.08 ± 0.045 | 0.03 ± 0.001 | 0.15 ± 0.001 | 0.04 ± 0.001 | 19.76** | 9.37** | 3.91* |
| DOC ($\mu g\,g^{-1}$) | 508 ± 53 | 817.4 ± 146 | 218.4 ± 26 | 170 ± 206.6 | 13.5 ± 14 | 239 ± 33 | 712 ± 132 | 371 ± 27 | 830 ± 375 | 94.3 ± 47 | 3.75ns | 4.66* | 0.36ns |
| DON ($\mu g\,g^{-1}$) | 4.6 ± 1 | 6.7 ± 0.9 | 2.1 ± 0.5 | 8.2 ± 1.1 | 8.2 ± 4.5 | 27 ± 1 | 10.3 ± 1.7 | 3.7 ± 0.6 | 25.5 ± 2.2 | 10.3 ± 0.6 | 13.09** | 12.72** | 4.47* |
| DIP ($\mu g\,g^{-1}$) | 0.02 ± 0.004 | 0.004 ± 0.004 | 0.02 ± 0.006 | 0.04 ± 0.006 | 0.005 ± 0.003 | 0.007 ± 0.004 | 0.01 ± 0.008 | 0.006 ± 0.006 | 0.14 ± 0.030 | 0.004 ± 0.04 | 8.95** | 24.81*** | 10.57*** |
| DOP ($\mu g\,g^{-1}$) | 1.6 ± 0.1 | 2.3 ± 0.2 | 1.8 ± 0.01 | 2.1 ± 0.6 | 1.8 ± 0.06 | 1.5 ± 0.05 | 1.5 ± 0.05 | 1.45 ± 0.05 | 0.65 ± 0.1 | 1.4 ± 0.04 | 1.33ns | 15.83** | 2.48ns |
| Ammonium ($\mu g\,g^{-1}$) | 0.4 ± 0.1 | 0.08 ± 0.08 | 2.9 ± 0.8 | 2 ± 0.9 | 1 ± 0.2 | 4.8 ± 0.2 | 4.3 ± 1.5 | 6.9 ± 1.2 | 7.8 ± 0.9 | 0.6 ± 0.1 | 29.08*** | 10.68** | 4.31* |
| Nitrates ($\mu g\,g^{-1}$) | 0 ± 0 | 0.6 ± 0.5 | 0.3 ± 0.2 | 1.1 ± 0.2 | 0.3 ± 0.2 | 1.4 ± 0.7 | 0 ± 0 | 0.2 ± 0.2 | 0 ± 0 | 0 ± 0 | 13.91** | 0.96ns | 1.61ns |
| micC ($\mu g\,g^{-1}$) | 0 ± 0 | 172.3 ± 45 | 267 ± 127 | 62 ± 10 | 92 ± 26 | 0 ± 0 | 182 ± 12 | 348.6 ± 69 | 139 ± 18 | 219 ± 38 | 10.27** | 6.22** | 0.89ns |
| micN ($\mu g\,g^{-1}$) | 0 ± 0 | 3.4 ± 0.6 | 1.6 ± 0.4 | 7.3 ± 0.2 | 2 ± 0.3 | 0 ± 0 | 3.2 ± 1 | 5.3 ± 1.4 | 6.8 ± 2.8 | 5.5 ± 0.5 | 2.89ns | 3.69* | 9.31** |

**Notes.**
* $P < 0.05$.
** $P < 0.01$.
*** $P < 0.0001$.
ns not significant.

**Table 2  Changes in biogeochemical parameters.** Biogeochemical parameters of microcosms and undisturbed soil samples of each site for the last sampling date, 12 months ($n = 3$). Mean $\pm$ SD with $F$ values and significance levels.

| | Dry Lagoon undisturbed | Dry Lagoon microcosms | River undisturbed | River microcosms | Treatment | Site | Site* treatment |
|---|---|---|---|---|---|---|---|
| | | | | | | **Effect** | |
| | | | | | | **$F$ values** | |
| TOC (mg.g$^{-1}$) | $2.9 \pm 0.3$ | $2.9 \pm 0.2$ | $23.2 \pm 3.9$ | $4 \pm 1.7$ | $19.7^{**}$ | $24.8^{**}$ | $19.9^{**}$ |
| TN (mg.g$^{-1}$) | $0.2 \pm 0.005$ | $0.24 \pm 0.015$ | $1.1 \pm 0.2$ | $0.7 \pm 0.06$ | $5.02^{ns}$ | $54.7^{***}$ | $6.6^{ns}$ |
| TP (mg.g$^{-1}$) | $0.023 \pm 0.001$ | $0.024 \pm 0.001$ | $0.06 \pm 0.008$ | $0.04 \pm 0.006$ | $2.3^{ns}$ | $26.9^{***}$ | $5.2^{ns}$ |
| DOC (μg.g$^{-1}$) | $13.5 \pm 9.4$ | $94.3 \pm 35.4$ | $20.4 \pm 9.4$ | $110 \pm 12.3$ | $0.2^{ns}$ | $38.8^{***}$ | $0.008^{ns}$ |
| DON (μg.g$^{-1}$) | $1.4 \pm 0.5$ | $8.2 \pm 4.5$ | $0.5 \pm 0.12$ | $9.5 \pm 0.6$ | $24.2^{***}$ | $0.15^{ns}$ | $2.8^{ns}$ |
| DIP (μg.g$^{-1}$) | $0.0 \pm 0$ | $0.005 \pm 0.003$ | $0.0 \pm 0$ | $0.004 \pm 0.004$ | $3.7^{ns}$ | $0.36^{ns}$ | $0.032^{ns}$ |
| DOP (μg.g$^{-1}$) | $1.5 \pm 0.04$ | $1.8 \pm 0.06$ | $1.1 \pm 0.05$ | $1.43 \pm 0.04$ | $45.5^{***}$ | $77.82^{***}$ | $0.2^{ns}$ |
| Ammonium (μg.g$^{-1}$) | $0.95 \pm 0.2$ | $1.3 \pm 0.1$ | $0.6 \pm 0.1$ | $1.45 \pm 0.1$ | $13.4^{**}$ | $0.26^{ns}$ | $3.06^{ns}$ |
| Nitrates (μg.g$^{-1}$) | $0.8 \pm 0.4$ | $0.34 \pm 0.2$ | $0.0 \pm 0$ | $0.0 \pm 0$ | $0.86^{ns}$ | $5.4^{*}$ | $0.85^{ns}$ |
| MicC (μg.g$^{-1}$) | $43.5 \pm 20$ | $92.1 \pm 25.7$ | $206 \pm 17$ | $219 \pm 38$ | $1.4^{ns}$ | $32^{***}$ | $0.46^{ns}$ |
| MicN (μg.g$^{-1}$) | $4.9 \pm 0.7$ | $2.1 \pm 0.3$ | $11.2 \pm 2.6$ | $5.5 \pm 0.5$ | $9.1^{*}$ | $12.3^{**}$ | $1.05^{ns}$ |

**Notes.**

$^{*}$ $P < 0.05$.

$^{**}$ $P < 0.01$.

$^{***}$ $P < 0.0001$.

$^{ns}$ not significant.

at 97% similarity were identified. The majority of sequences (>80%) belonged to one of the nine major phyla (arranged in order of abundance): Acidobacteria, Proteobacteria, Bacteroidetes, Firmicutes, Chloroflexi, Deinococcus-Thermus, Gemmatimonadetes, Actinobacteria, or Candidatus division TM6 (Table 3, Figure S1). A higher number of unclassified sequences were observed at the Dry Lagoon site than in the River site. Interestingly, none of the sequences belonged to any eukaryotic specie which mitochondria and chloroplast would be amplified by these techniques, suggesting that wind and water could help the colonization processes. We believe that nematodes and other invertebrates did not play an important role in the migration of bacteria between disturbed and undisturbed sites.

Despite the differences observed in nutrient content between sites, the 16S rRNA sequence analysis showed similar variations in the diversity and community composition of the sites, even though the overall diversity was higher in the River during the entire sampling period (Table 4, Fig. 1). At both sites, during the first six months (3 and 6 months samples) there was an increase in diversity, then the diversity decreased by nine months and increased again after one year (12 months) following a similar fluctuating pattern to the biomass. Contrasting the diversity in the last sampling date with the diversity of the adjacent undisturbed soil, we observed that the undisturbed soil had higher diversity than microcosms samples (Table 4).

Prior to disturbance, both Dry Lagoon and River communities were dominated by Acidobacteria (44% and 60%, respectively) (Table 3). The Dry Lagoon had a large

**Table 3 Microbial diversity.** Distribution of bacterial phyla. The relative abundance of bacterial phyla in each of the sampling dates and site. Values represent the percentage of each group in the respective library.

| | Dry lagoon | | | | | | River | | | | | |
|---|---|---|---|---|---|---|---|---|---|---|---|---|
| | Before sterilization | 3 months | 6 months | 9 months | 12 months | Undisturbed soil | Before sterilization | 3 months | 6 months | 9 months | 12 months | Undisturbed soil |
| Acidobacteria | 44.41 | 30.56 | 42.02 | 24.26 | 42.68 | 33.36 | 59.60 | 49.32 | 47.49 | 39.04 | 46.49 | 48.55 |
| Bacteroidetes | 4.68 | 0.24 | 2.40 | 7.40 | 7.34 | 3.16 | 9.03 | 0.00 | 5.35 | 9.28 | 8.07 | 3.40 |
| Chloroflexi | 0.98 | 0.01 | 1.25 | 4.61 | 0.97 | 9.36 | 2.05 | 8.37 | 2.85 | 4.90 | 3.73 | 11.72 |
| Gemmatimonadetes | 0.70 | 0.02 | 1.71 | 0.82 | 1.48 | 0.48 | 1.92 | 0.02 | 3.25 | 3.17 | 2.00 | 1.40 |
| Deferribacteres | 0.00 | 0.00 | 0.00 | 0.00 | 0.00 | 0.02 | 0.90 | 1.08 | 1.62 | 0.26 | 0.37 | 1.60 |
| TM6 | 0.46 | 0.00 | 0.13 | 4.53 | 0.51 | 0.80 | 0.89 | 0.02 | 0.94 | 3.60 | 0.38 | 1.09 |
| Firmicutes | 0.31 | 10.73 | 0.59 | 2.04 | 0.37 | 2.44 | 0.82 | 21.69 | 0.86 | 2.16 | 0.98 | 2.80 |
| Actinobacteria | 0.63 | 4.69 | 1.00 | 0.74 | 0.43 | 1.74 | 0.60 | 0.19 | 1.24 | 0.71 | 0.73 | 1.65 |
| Deinococcus–Thermus | 2.80 | 0.33 | 6.09 | 2.26 | 0.69 | 1.21 | 0.45 | 0.00 | 6.27 | 5.62 | 1.02 | 0.46 |
| Cyanobacteria | 0.20 | 0.31 | 0.20 | 0.06 | 0.21 | 0.46 | 0.39 | 0.16 | 0.55 | 0.27 | 0.33 | 0.48 |
| Verrucomicrobia | 0.08 | 0.00 | 0.22 | 2.38 | 1.24 | 0.72 | 0.31 | 0.00 | 0.50 | 1.30 | 1.34 | 0.91 |
| Chlamydiae | 1.01 | 0.01 | 0.38 | 0.29 | 0.85 | 1.62 | 0.54 | 0.00 | 0.34 | 1.28 | 0.86 | 1.81 |
| Planctomycetes | 0.51 | 0.00 | 0.20 | 0.08 | 0.01 | 0.03 | 0.12 | 4.62 | 1.22 | 1.65 | 1.38 | 0.38 |
| TM7 | 0.23 | 0.00 | 0.05 | 3.65 | 0.42 | 0.22 | 0.39 | 0.00 | 0.39 | 1.18 | 0.52 | 0.43 |
| TA06 | 0.00 | 0.00 | 0.00 | 0.00 | 0.00 | 0.00 | 0.00 | 2.97 | 0.08 | 0.00 | 0.51 | 0.06 |
| OP11 | 0.00 | 0.00 | 0.00 | 0.00 | 0.00 | 0.00 | 0.02 | 0.00 | 0.41 | 1.86 | 0.02 | 0.82 |
| unclassified | 33.59 | 12.01 | 33.56 | 26.63 | 18.21 | 30.55 | 0.00 | 0.00 | 0.00 | 0.00 | 0.00 | 0.00 |
| *Proteobacteria* | | | | | | | | | | | | |
| Alphaproteobacteria | 1.88 | 0.16 | 1.14 | 3.52 | 4.32 | 2.47 | 7.23 | 0.15 | 4.37 | 3.39 | 5.75 | 4.24 |
| Betaproteobacteria | 0.04 | 6.56 | 0.03 | 0.41 | 0.40 | 0.03 | 1.35 | 0.01 | 0.19 | 0.04 | 0.53 | 0.19 |
| Deltaproteobacteria | 0.54 | 8.95 | 0.30 | 1.04 | 0.59 | 3.64 | 4.31 | 0.85 | 4.98 | 3.77 | 2.01 | 8.59 |
| Gammaproteobacteria | 5.84 | 24.95 | 7.32 | 12.27 | 17.47 | 3.86 | 21.40 | 10.52 | 15.20 | 13.36 | 20.26 | 5.27 |
| Other groups (<1%) | 1.10 | 0.47 | 1.41 | 3.02 | 1.80 | 3.86 | 1.70 | 0.04 | 1.92 | 3.17 | 2.69 | 4.15 |

**Table 4 16S diversity.** Bacterial 16S rRNA amplicon diversity analysis of two arid soil sites in Cuatro Cienegas, Coahuila, Mexico.

| | OTUs[a] | Singlets[a] | Doublets[a] | Chao1 | CI[b] | Shannon | CI[b] | Simpson | CI[b] | Ace | CI[b] | % coverage[c] |
|---|---|---|---|---|---|---|---|---|---|---|---|---|
| **Dry lagoon** | | | | | | | | | | | | |
| *Before sterilization* | 6586 | 4917 | 882 | 43226 | (39437–47450) | 7.31 | (7.27–7.34) | 0.007 | (0.006–0.007) | 99811 | (96592–103144) | 0.63 |
| *3 months* | 4106 | 2760 | 587 | 20166 | (18253–22337) | 6.07 | (6.03–6.11) | 0.020 | (0.019–0.021) | 47720 | (45885–49636) | 0.78 |
| *6 months* | 7518 | 5830 | 1093 | 42815 | (39583–46372) | 7.59 | (7.56–7.63) | 0.004 | (0.004–0.004) | 106755 | (103366–110262) | 0.61 |
| *9 months* | 3552 | 2433 | 451 | 10097 | (9334–10961) | 7.17 | (7.15–7.21) | 0.002 | (0.002–0.003) | 18720 | (17980–19497) | 0.77 |
| *12 months* | 6882 | 5114 | 808 | 60661 | (54582–67515) | 7.70 | (7.66–7.73) | 0.004 | (0.004–0.005) | 137126 | (133112–141267) | 0.58 |
| *Undisturbed soil* | 8258 | 7157 | 1135 | 53773 | (49545–58434) | 8.16 | (8.13–8.19) | 0.001 | (0.001–0.001) | 130419 | (126816–134131) | 0.57 |
| **River** | | | | | | | | | | | | |
| *Before sterilization* | 8268 | 6797 | 1085 | 53676 | (49534–58233) | 7.80 | (7.76–7.83) | 0.005 | (0.004–0.005) | 140680 | (136580–144910) | 0.56 |
| *3 months* | 3785 | 2878 | 657 | 17395 | (15706–19323) | 6.04 | (6.00–6.08) | 0.013 | (0.013–0.014) | 32413 | (31077–33815) | 0.81 |
| *6 months* | 9001 | 7289 | 1148 | 61455 | (56784–66583) | 8.24 | (8.21–8.26) | 0.001 | (0.001–0.002) | 161368 | (156754–166125) | 0.52 |
| *9 months* | 6530 | 6607 | 1202 | 33236 | (30685–36056) | 7.35 | (7.32–7.38) | 0.003 | (0.003–0.003) | 94400 | (91445–97457) | 0.69 |
| *12 months* | 6356 | 4728 | 734 | 27928 | (25926–30135) | 7.80 | (7.75–7.82) | 0.002 | (0.002–0.002) | 67224 | (65075–69452) | 0.62 |
| *Undisturbed soil* | 8461 | 7019 | 1171 | 52934 | (48931–57332) | 8.23 | (8.20–8.25) | 0.001 | (0.001–0.001) | 125851 | (122223–129596) | 0.54 |

**Notes.**

All samples were normalized to 10,441 sequences (Table S3) for diversity comparison, values in parentheses represent the lower and upper 95% confidence interval associated with the Chao1 nonparametric estimator.

[a] Values calculated based on 97% threshold.

[b] Lower and upper 95% confident intervals associated with the diversity parameter.

[c] Good's coverage estimator.

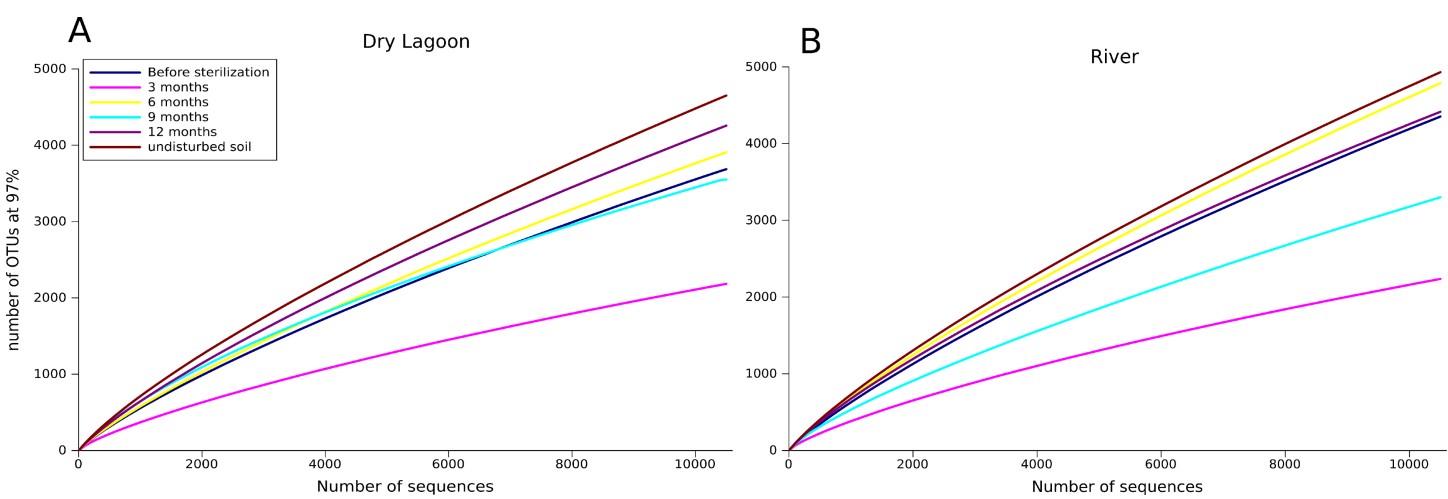

**Figure 1 Sampling effect.** Rarefaction curves of the A) Dry Lagoon and B) River in all sampling dates (3, 6, 9 and 12 months). OTUs were determined at 97% sequence identity.

proportion of bacteria that were unclassified at the 97% identity threshold (12 to 33%). Proteobacteria were also abundant, comprising 9% of the Dry Lagoon sequences and 35% of the River sequences. Overall there was a gradual increase in the abundance of the Bacteroidetes during the succession process, which were not abundant in the first sampling (less than 0.24%) but increased with time (around 7–9%). Autotrophic phyla such as Clorobi, Chloroflexi and Cyanobacteria, increased in abundance the first nine months and then decrease their abundance in the last sampling (12 months) (Table 3).

A more detailed analysis at the family level showed that the families present before the sterilization in low abundance were not found after 3 months, but many of them were recovered by 6 months and onward (Table S4). The initial colonizers at 3 months were different between sites, but in both cases, the majority of the identified families were related to known opportunistic heterotrophs. In the Dry Lagoon the most abundant families were Moracellacea, Streptococcaceae, Alcaligenaceae, Pseudomonadaceae, Micrococcaceae and Bacillaceae. At the River site at 3 months, Holophagaceae, Acidobacteriaceae, GIF3 (Chloroflexi), Enterobacteriaceae, Streptococcaceae, Listeriaceae and Bacillaceae were found. By 6 months the abundance of unclassified taxa increased, but also the abundance of families that include previously described members with other metabolic capabilities besides heterotrophy (Table S4). The relative abundances of some bacterial groups at different taxonomic levels correlated significantly with Total Organic C, Total N, humidity and $NO_3^-$ (Table 5), and these trends were corroborated with the CCA (Fig. 2).

Similarity between sites and samples through the time, measured with the Bray-Curtis distance and Jaccard coefficient using OTUs at 97% of identity, achieved a peak three months after sterilization, while it was lower before sterilization and six, and nine months after sterilization. After a year the microcosms communities were more similar to the communities from undisturbed soil at each site (Fig. 3).

**Table 5 Relationship between soil and diversity.** Spearman's rank correlations between the relative abundances of the most abundant bacterial phyla, proteobacterial classes and bacterial families, and the soil properties in CCB. Bold numbers: $P < 0.05$; Bold and underlined numbers $P < 0.001$.

| Taxonomic group | Humidity | Total organic carbon | Total nitrogen | Nitrate |
|---|---|---|---|---|
| **Phyla** | | | | |
| Acidobacteria | 0.54 | **0.59** | **0.57** | **−0.66** |
| Bacteroidetes | 0.46 | 0.46 | 0.51 | −0.16 |
| Chloroflexi | **0.71** | **0.65** | 0.51 | −0.24 |
| Firmicutes | 0.24 | 0.27 | 0.26 | −0.01 |
| Deinococcus-Thermus | 0.02 | −0.13 | −0.23 | 0.05 |
| Gemmatimonadetes | 0.50 | 0.43 | 0.51 | −0.27 |
| Actinobacteria | −0.07 | −0.30 | −0.16 | 0.42 |
| TM6 | **0.58** | **0.59** | 0.49 | 0.03 |
| Cyanobacteria | 0.50 | 0.32 | 0.50 | −0.12 |
| **Class** | | | | |
| Alphaproteobacteria | 0.49 | 0.47 | **0.64** | −0.10 |
| Betaproteobacteria | −0.06 | −0.02 | 0.24 | 0.22 |
| Deltaproteobacteria | 0.44 | 0.37 | 0.53 | −0.01 |
| Gammaproteobacteria | −0.12 | −0.02 | 0.22 | 0.07 |
| **Family** | | | | |
| Acidobacteriaceae | **0.69** | **0.62** | **0.63** | **−0.60** |
| Holophagaceae | **0.75** | **0.84** | **0.78** | **−0.57** |
| RB25 (Holophagae) | **0.71** | **0.68** | **0.71** | **−0.65** |
| SJA-36 (Holophagae) | **0.67** | **0.74** | **0.71** | −0.70 |
| Unclassified Acidobacteria | **−0.93** | **−0.89** | **−0.82** | **0.60** |
| Moraxellaceae | −0.12 | −0.22 | −0.29 | −0.05 |
| Chitinophagaceae | 0.42 | 0.41 | 0.53 | −0.04 |
| Truepera | −0.11 | −0.24 | −0.30 | 0.07 |
| Anaerolineaceae | 0.63 | 0.47 | 0.44 | 0.09 |
| Pseudomonadaceae | −0.45 | −0.24 | −0.10 | 0.18 |
| Streptococcaceae | −0.50 | −0.43 | −0.53 | 0.24 |
| Enterobacteriaceae | **0.56** | 0.51 | 0.39 | **−0.84** |
| GIF3 (Chloroflexi) | **0.76** | **0.79** | **0.76** | **−0.73** |
| Alcaligenaceae | −0.37 | −0.24 | −0.03 | 0.50 |
| unclassified Deltaproteobacteria | **−0.80** | **−0.85** | **−0.82** | **0.66** |
| Listeriaceae | 0.15 | 0.20 | 0.18 | −0.46 |

## DISCUSSION

*Microbial diversity and secondary succession in a gypsum based soil.* In this study we analyzed the bacterial secondary succession of experimentally disturbed soil microcosms at two sites over one year. While each site differed in both biogeochemistry and biodiversity, both present similar community composition at the beginning of the experiment. However, successional patterns differed in later stages. This suggests an species sorting effect at the beginning of the experiment, followed by a more neutral process in later stages. At the end of the experiment, comparing the adjacent soil samples with the

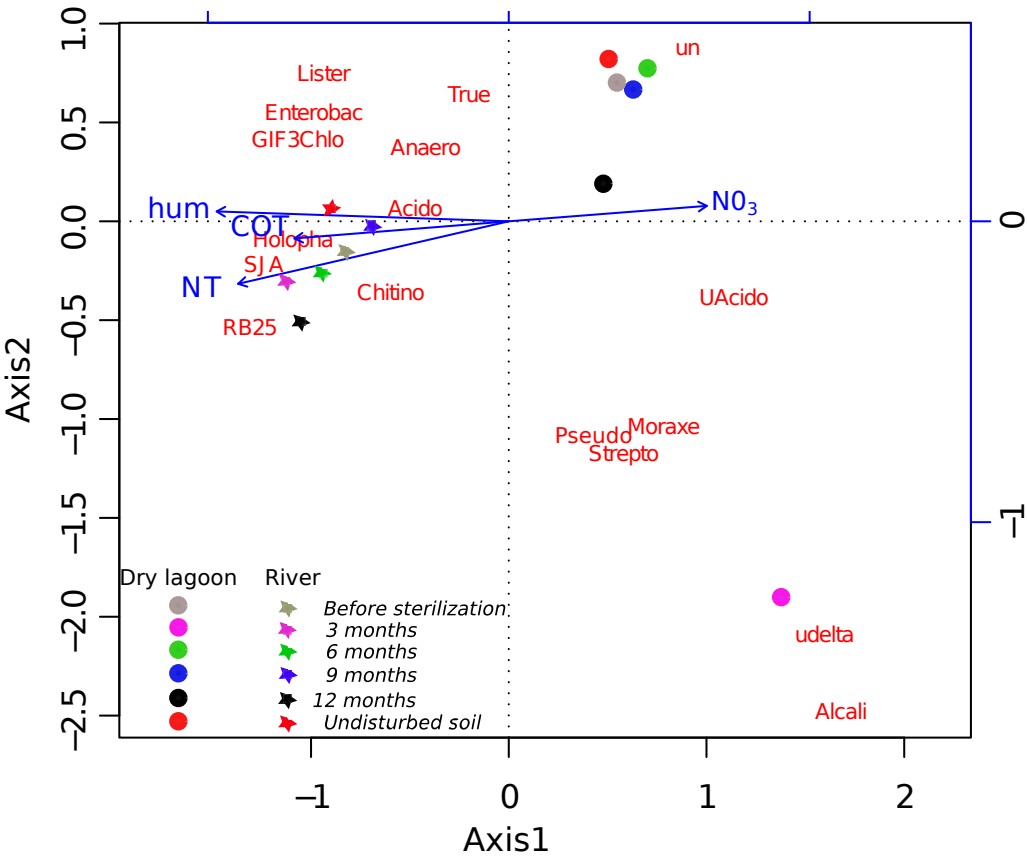

**Figure 2  Relationship between diversity and environment.** CCA ordination biplot of the most abundant family composition during the secondary succesional process. Vectors represent the soil characteristics, while centroids indicate different 454 pyrosequence 16S rRNA gene amplicon libraries across the time. Temporal replicates that are close together on the ordination diagram are more similar (in terms of their microbial community structure) than replicates that are farther apart.

experimentally disturbed soil, the microbial communities at either site do not resemble adjacent undisturbed microbial communities. Both disturbed soils were most closely related to each other (Fig. 3).

Four factors contribute strongly to differences in our sites: humidity, Total N, Total Organic C and Total P. The first three factors were identified as the most correlated variables with the relative abundance of the bacterial groups. These factors have been considered as part of the best group of soil variables for predicting microbial community composition in arid lands (*Collins et al., 2008*; *Wall & Virginia, 1999*). We also found that Total P in the soil explained some variability. This was expected, as P is the most limiting nutrient in CCB (*Elser et al., 2005*).

In a previous 16S rRNA Sanger-based clone library analysis (*López-Lozano et al., 2012*), 15 phyla and 40 classes were identified from 293 sequences in the harsher Dry Lagoon. The River site had 16 phyla and 36 classes identified from 223 sequences. Even with the higher sequencing effort afforded by 454 pyrosequencing, we still only captured 52% to 81% of the microbial communities in our samples (Fig. 1 and Table 4). This was

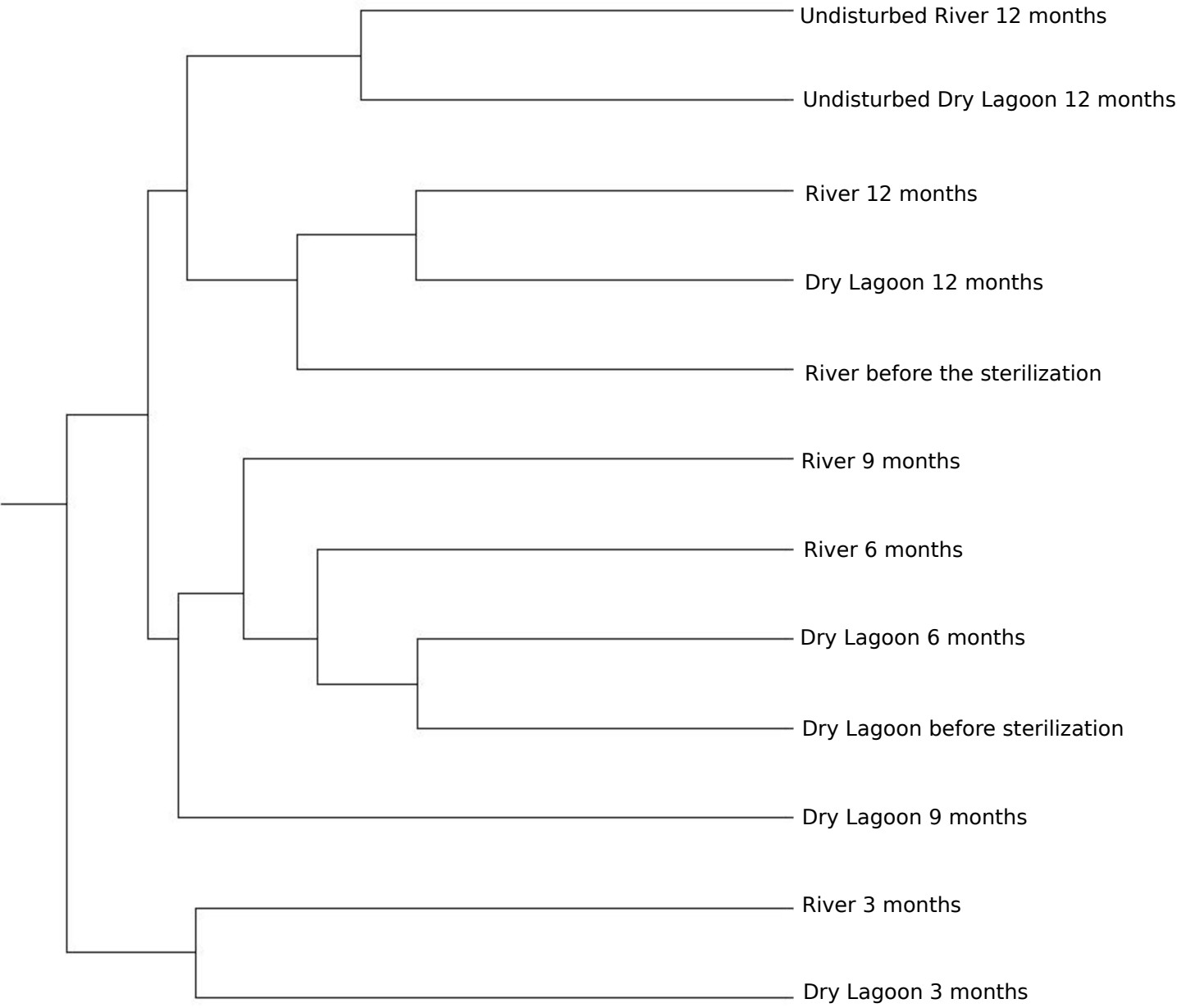

**Figure 3  Relationship between samples.** Clustering of the 16S rRNA community composition at 97% similarity based on the Bray-Curtis algorithm of ecotype abundance. The cluster diagram of sample similarity generated using the Jaccard Index showed the same pattern (data not shown).

surprising, since in more fertile and wet soils analyzed with similar sequencing effort, the maximum number of OTUs at 97% of identity seldom exceeds ∼5,600 OTUs (*Roesch et al., 2007*), while in our more diverse sample, we estimated ∼9,000 OTUs at 97% identity. However, the sites analyzed in this study differed in diversity, with soil adjacent to the River being more diverse than the Dry Lagoon if we compare the OTUs at 97% (Table 4). In the previous clone library study the same pattern was observed (*López-Lozano et al., 2012*). We also evaluated the community composition in terms of "ecological groups",

considering groups in which abundance correlated with high or low C mineralization rates (copiotrophs or oligotrophs, respectively) in general soil surveys (*Fierer, Bradford & Jackson, 2007*; *Fierer et al., 2010*), and widely known autotrophs (phototrophs such as Cyanobacteria and Chloroflexi). Also we determined some metabolic capabilities by comparison of the sequences with the closer cultivated organisms. Despite the broad ecological classification based only in 16S rRNA gene sequences, our results showed interesting trends in groups considered as oligotrophs, copiotrophs (organism that need more nutrients) as well as heterotrophs and autotrophs.

The 16S rRNA libraries of both sites are dominated by Acidobacteria in all samples. This is one of the most common phyla found in soil libraries worldwide (*Janssen, 2006*). The Acidobacteria are in general oligotrophic (*Eichorst, Breznak & Schmidt, 2007*; *Fierer, Bradford & Jackson, 2007*) and have been shown to comprise 50% of clone libraries in arid soils (*Dunbar et al., 1999*; *Kuske, Barns & Busch, 1997*), but they are less abundant in more nutrient-rich agricultural soils (*Nagy, Pérez & Garcia-Pichel, 2005*; *Roesch et al., 2007*). Hence, it is not surprising that they represent ∼30–60% of bacteria in all our libraries in CCB sampling sites. In contrast, Bacteroidetes have been classified as copiotrophs in general (*Fierer, Bradford & Jackson, 2007*). In our libraries Bacteroidetes become much more abundant at the 6 months sampling, and it is possible that these phyla appear when the accumulation of organic material is enough to sustain the mineralization rate of this group. In fact, between 6 and 9 months there was a drop in the Dissolved Organic C while the microbial C increase in the soil microcosms. Autotrophic groups, such as Cyanobacteria are abundant at both sites. This phylum is common in most environments including soil and biological soil crust of arid zones (*Gundlapally & Garcia-Pichel, 2006*; *Nagy, Pérez & Garcia-Pichel, 2005*). The abundance of yet another autotrophic group, the Chloroflexi, increased during the initial stages of succession (3–9 months) but decreased in the last sampling date (12 months). With the exception of high Chloroflexi in the River sample during the first samplings, the majority of the colonizers are opportunistic heterotrophs (a greater part members of Firmicuttes, Betaproteobacteria and Gammaproteobacteria). It is possible that especially in the Dry Lagoon site, the initial colonizers were benefited by the Dissolved Organic C, released during the sterilization process. The nutrient pulse could have facilitated colonization by heterotrophs that benefited from available resources in the disrupted microcosm. The finding of preferential growth of opportunistic heterotrophs during early succession agrees with the findings from another study using outdoor sterile microcosms seeded by rainwater bacteria (*Langenheder & Székely, 2011*). These authors found that neutral and species sorting processes interacted during the assembly of bacterial communities, and the importance of each depended on how many generalists and specialists were present in the community. We suggest that in our microcosms, the initial faster growing community (generalists) depleted the nutrients, and then were out competed by groups that have alternative energy sources (specialists). At 6 months the abundance of groups with more specialized metabolic capabilities increased, and the taxa with low abundances before the sterilization, "the rare biota" that were not present after three months returned to the community from

this sampling date. The observed nitrate accumulation in the Dry Lagoon site could be due to an increase in nitrification, an alternative mechanism of obtaining energy that could persist under those conditions (*Montaño, García-Oliva & Jaramillo, 2007*). Also, the higher DOP in the Dry Lagoon site suggests that P mineralization is not occurring as efficiently as in the River site, perhaps because of limited energy for exo-enzyme production necessary for acquisition (F Garcia-Oliva, personal communication). Hence, the Dry Lagoon site might be supporting a more oligotrophic microbiota than the River site. Conversely, changes in DOC at the River site correlated with changes in microbial C. Additionally, ammonium correlated with microbial N, suggesting that the initial surge of nutrients could have facilitated a faster migration of the neighboring heterotrophic biomass in this site than in the Dry Lagoon site. Photosynthetic autotrophs (such as Chloroflexi) seem to be sufficient for nurturing the community in the River site providing C sources, since nitrates and DOP are scarce, indicating low nitrification rates.

Regarding the diversity shifts, both sites showed similar patterns. There is an initial increase in the Shannon index, followed by a decrease at 9 months and an increase at 12 months. We do not discard the possibility that these convergent patterns only in the diversity index shifts can be related to climatic factors. At nine months a decrease in diversity was observed, which coincided with an unusually heavy precipitation event (Table S5).

The Bray-Curtis similarity index and Jaccard similarity coefficient analyses, suggest that even though the initial communities are similar in composition, they differ in later stages. In addition, the analysis by phylogenetic and ecological groups, suggest that the communities respond in a similar trajectory of initial colonization first by heterotropic generalists and later specialists. However the River site was characterized by higher nutrients and diversity, together with the continuous presence of Cloroflexi in high abundance suggest an early food web based on primary production. This pattern is similar to the early stages of the autotrophic succession suggested by *Fierer et al. (2010)*. Our experimental design cannot be used to test this autotrophic model, but our data show that the colonizers were a mix of heterotrophs and autotrophs.

In this study we found community composition of the soil microcosms after perturbation did not recover to similar communities found in the undisturbed soil community after one year. We are not assuming that there is only one "climax" community in terms of composition. What we suggest is that a community would be "recovered" when it has similar performing conditions to the neighboring undisturbed community. The performance of the community can be inferred by the soil nutrient content and characteristics. Due to the significant differences in physicochemical parameters between microcosms and control sites at 12 months, we can conclude that despite the diversity found at the Churince soils, the small patches (1 kg mesh bags) did not recover to resemble undisturbed soil in either site, after a year of migration and succession. Nevertheless, it seems they followed a parallel paths for such recovery at the beginning of the succession and diverged in later stages.

## CONCLUSIONS

This is the first study on bacterial soil dynamics conducted at the CCB. Our results provide important insight into microbial community dynamics in response to a disturbance (secondary succession). In this study we found changes in community composition across time that were indicative of the successional process. Our data showed evidence of similar initial colonizers in both sites, but the divergence in later successional stages reflects stochastic factors suggesting a species sorting and neutral effects. While community descriptions based only on 16S rRNA gene analysis do not reflect the full metabolic plasticity of the communities, this study provides important insight into microbial succession patterns. For further research metagenomics and measures of the rates of the physiological traits are necessary to corroborate our results. Notwithstanding, it is interesting that succession of small, perturbed spots is very slow; this reveals the temporal scale is important for this community in terms of resilience, but general long-term monitoring is necessary to better understand the temporal patterns and natural variability of this area. Changes in microbial communities due to disturbance may directly affect ecosystem processes, which are vital in a protected natural area, threatened by over-exploitation of aquifers such as is occurring in the CCB.

## ACKNOWLEDGEMENTS

We are grateful to Rodrigo González Chauvet, German Bonilla Rosso and other members within the laboratory of Evolución Molecular y Experimental for assistance with the sample collection; to Celeste Martínez-Piedragil, Rodrigo Velázquez-Durán and Maribel Nava-Mendoza for assistance with soil chemical analysis; to Laura Espinosa Asuar and Erika Aguirre Planter for help with molecular, sequencing work and logistic support; to Ana Gutiérrez-Preciado and Jaime Gasca-Pineda for bioinformatics assistance. We specially thank the APFF of Cuatro Cienegas for their support and logistics.

### Funding

This study was funded by grants from CONACyT-Semarnat 2006-C01-23459 and WWF-Alianza Carlos Slim OL039 to Valeria Souza and Posgrado en Ciencias Biomédicas and CONACyT (grant No 44810) to Nguyen E. López-Lozano. The funders had no role in study design, data collection and analysis, decision to publish, or preparation of the manuscript.

### Grant Disclosures

The following grant information was disclosed by the authors:
CONACyT-Semarnat: 2006-C01-23459.
WWF-Alianza Carlos Slim: OL039.
Posgrado en Ciencias Biomédicas.
CONACyT: 44810.

## Competing Interests

Luis Eguiarte and Valeria Souza are Academic Editors for PeerJ.

## Author Contributions

- Nguyen E. López-Lozano, Luis E. Eguiarte and Valeria Souza conceived and designed the experiments, performed the experiments, analyzed the data, contributed reagents/materials/analysis tools, wrote the paper.
- Karla B. Heidelberg performed the experiments, analyzed the data, contributed reagents/materials/analysis tools, wrote the paper.
- William C. Nelson performed the experiments, analyzed the data, contributed reagents/materials/analysis tools, genebank accesions.
- Felipe García-Oliva conceived and designed the experiments, performed the experiments, analyzed the data, contributed reagents/materials/analysis tools.

## Field Study Permissions

The following information was supplied relating to ethical approvals (i.e. approving body and any reference numbers):

Collecting permit Vida Silvestre FAUT230, Mexico to VS.

## DNA Deposition

The following information was supplied regarding the deposition of DNA sequences:

The GenBank accession numbers for 16S rRNA pyrosequences in this work are in the BioProject: PRJNA167137.

## Data Deposition

The following information was supplied regarding the deposition of related data:

CONABIO database, work in progress.

## Supplemental Information

Supplemental information for this article can be found online at http://dx.doi.org/10.7717/peerj.47.

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
