# Peer review of "Microbial secondary succession in soil microcosms of a desert oasis in the Cuatro Cienegas Basin, Mexico"

_PeerJ, doi:10.7717/peerj.47_

## Round 0.1 · original submission · Minor Revisions

Thank you for an interesting and generally well-conceived and written manuscript, which should be suitable for publication with minor revisions. Please address the reviewer's comments and criticisms in detail.

Reviewer 1 ·

Basic reporting

refer to general comments, where I have included some specific editing requests in the text and figures

Experimental design

refer to general comments, where I have requested more clarity in methods

Validity of the findings

refer to general comments, where I have requested more discussion on the CCA

Additional comments

This paper investigates microbial secondary succession in a desert. The science is sound, the background in the introduction is very good, although the objectives need to be more clearly stated. There is an attempt to tie these results to the discussion of species sorting vs neutral processes as controls on microbial community structuring, which is great – ecological theory in microbial ecology is wonderful! The authors present informed opinions, not hard conclusions, on what theories are controlling their results, which is appropriate for a field study. However, the theory should be stated more clearly in the objectives and in the outline of the discussion, to balance the story introduced in the introduction. The methods could use a few more details and the results need some editing but are otherwise clear. I would like to know why the authors chose to use a CCA instead of a (or perhaps in addition to) a less constrained ordination of community data and environmental data, to see the full community gradient and an independent assessment of the importance of these variables. I enjoyed the discussion. This paper is ready to publish with a few edits, comments and edits below by line number.

76-82. As written, the two objectives are not clearly distinguished from each other. Perhaps the first is to evaluate the processes (species sorting, stochastic) that drive microbial diversity, etc...the second is the extent of recovery? The last sentence is unnecessary.
106-109. Either the methods are not clearly describing the number of mesocosm bags sampled (3 reps x 4 dates = 12 per site), or there are extra bags that were not used. Please clarify. If not used for this experiment, perhaps say so or assign them to another experiment, whatever actually happened.
113-114. Biogeochemistry and MB sensitive to storage time – please say how long they were stored for.
120. Which standard procedures? These references all contain multiple assays, please be more specific as to which assay you ran.
152-199. Move up to before stats in methods.
196-199. A very brief explanation of what these statistics measure and how they differ is needed here.
296-297. The sentence proposing a species sorting effect is hanging here. It would be much more appropriate followed by a supporting discussion – perhaps as the first sentence in the second paragraph
303-309. This paragraph is weak; it repeats partial sentences (1st and 3rd sentence), and repeats concepts but using different terms for microbial community structure (2nd and last sentence). This paragraph is an ideal place to respond to your objectives – I realize this is a style comment, but it would certainly improve clarity of the story if you defend species sorting strongly early in your discussion.
328-329. Weak ending sentence. ‘Interesting trends’ should be replaced with a more specific result that segues to next paragraph.

Tables/figures comments:
Table 1 – bad formatting in first row. Move ‘river’, ‘site’ and ‘time’ column headings over; if numbers must be wrapped, please do this at the +/- sign, not in the middle of a number
Table 2 – are the site and treatment F-scores reversed for DOC?
Table 3 – bad formatting in first row. Move divider between sites over.
Table 5 – spell out COT and NT
Figure 2 – Nutrientes = nutrients; As mentioned earlier, the CCA is a constrained ordination - it ignores community structure that is not related to environmental variables. Therefore the text supporting this figure is misleading, it should emphasize that this is the community structure that has been constrained by the environmental variable matrix. Otherwise readers will try to relate this community structure – which is very tight within site as constrained by biogeochemistry, etc – to the cluster analysis, which is tighter within time in some cases, and by site in others. I would prefer it if the authors chose an unconstrained ordination on the community data (such as a NMDS), and then related that ordination to environmental variables to allow an expression and independent measure of the importance of pure community gradients. The authors may have chosen the CCA for a particular theoretical reason – if so, they should defend this in their methods and discuss the implications of this method of analysis in their discussion.

Typos/grammar:
49. resul = result
55. trough the time = through time
70. despite been highly diverse = despite the high diversity
72. evaluate = to evaluate
73. facing = after
74. gama = gamma
92. separate = separated
186. calculated = calculate
216. insert ‘and’ before ‘in comparison’
218. the time = time
226. higher values at 6 months, after that decreased gradually = higher values at 6 months, and then decreased gradually
352. of = from
381-382. At nine months, when a decrease in diversity was observed coincided with an… = At nine months a decrease in diversity was observed, which coincided with an…
389. generalist = generalists; specialist = specialists
390. as well as for = together with
405. did not recover to resemble undisturbed soil after a year of migration and succession in neither site = did not recover to resemble undisturbed soil in either site, after a year of migration and succession.
420. slow, = slow;

Reviewer 2 ·

Basic reporting

The paper appropriately sets the scene for the study undertaken and its importance. The reporting is well structured making good use of English in the main although there should be a thorough reread and amendments made to methodology text where there is some difficulty in understanding work done.
The work and related research is well introduced and appropriately referenced throughout.
Main problems relate to the tables and Figures. The Tables are disappointingly presented with poor distribution of data and headings. The legends should be examined for clarity. Table 3 has much confusing data and I wonder if there is a better way of presenting numbers that vary significantly with no clear trends. The variability with data of low abundance is a function of the analysis. Perhaps there should only be presentation of the phyla with high abundance with a list of low abundance phyla catalogued in the text with associated statistics as required.
The figures in colour are confusing and the use of symbols to differentiate the lines is recommended.

Experimental design

The methodology is inadequately described and presented. For example, where it states “All carbon forms were determined with….”, does this mean organic and inorganic carbon as total carbon without differentiation. There needs to be clarity as a TOC analyser does not differentiate all carbon forms. Similarly, “nitrogen (N) and phosphorus (P) forms were determined using a Bran-Luebbe Auto Analyzer (III)…” Here the authors need to clarify the forms of the elements and how these were prepared for autoanalysis.
How can one extract inorganic N (NH4+ and NO3) and inorganic dissolved nutrients using a spectrophotometric method? Sense? More details are needed. Was fixed and exchangeable NH4 extracted and determined? More needed on this important methodology.
The statistical and molecular aspects of the study were well considered and methodology appropriate.
The use of microcosms of soil in permeable bags may be a problem with respect to movement of larger organisms such as earthworms and nematodes that without doubt would affect decolonization of the disturbed environment and no consideration of this is evident.

Validity of the findings

The major finding is that the recovery of the disturbed soils is slow. It is obvious that oligotrophy and lack of water affects microbial recovery as in the former biomass development is affected and in the latter microbial mobility in dry soil is limited. Thus it is not surprising that recruitment is slow. Recrutiment may be faster if the systems had no restriction to the movement of larger organisms that can transport populations within the soil, eg. Earthworms and nematodes etc.

Additional comments

Targeted additional information on methodology, discussion of the role of other macrorganisms in soil recolonization and better presentation of Table and Figures represent minor changes to be made for acceptance of the interesting work. Longer term study needed for these oligotrophic and arid environments.

---

## Round 0.2 · accepted · Accept

Thank you for addressing the reviewers' comments so comprehensively - the changes have greatly enhanced the manuscript. Congratulations on being among PeerJ's earliest authors.